# Improvement of the Integral Indicator of the Ecological and Toxicological Assessment of the Danger of the Use of Pesticides in Agriculture and the Creation of an Information System for Their Monitoring

Irina Slastya [1], Elena Khudyakova [2,*], Ivan Vasenev [1], Dmitrii Benin [3], Marina Stepantsevich [2], Vladimir Vodjannikov [4], Mikhail Nikanorov [2] and Tatiana Fomina [5]

1 Department of Ecology, Institute of Amelioration, Water Management and Construction, Russian State Agrarian University—Moscow Timiryazev Agricultural Academy, Moscow 127550, Russia; slastya@rgau-msha.ru (I.S.); vasenev@rgau-msha.ru (I.V.)

2 Department of Applied Information Science, Institute of Economics and Management in Agribusiness, Russian State Agrarian University—Moscow Timiryazev Agricultural Academy, Moscow 127550, Russia; stepancevich@rgau-msha.ru (M.S.); nikanorov@rgau-msha.ru (M.N.)

3 Department of Agricultural Water Supply, Sanitation, Pumps and Pumping Stations, Russian State Agrarian University—Moscow Timiryazev Agricultural Academy, Moscow 127550, Russia; dbenin@rgau-msha.ru

4 Department of Production Organization, Russian State Agrarian University—Moscow Timiryazev Agricultural Academy, Moscow 127550, Russia; vtvodyannikov@rgau-msha.ru

5 Department of Foreign and Russian Languages, Russian State Agrarian University—Moscow Timiryazev Agricultural Academy, Moscow 127550, Russia; t.fomina@rgau-msha.ru

* Correspondence: evhudyakova@rgau-msha.ru; Tel.: +7-9165185228

**Abstract:** The aim of the work was to assess the ecological and toxicological dangers of pesticides using the value of a complex indicator determined by the calculation method. An integral indicator of the relative ecological and toxicological danger of pesticide use ($H_r$) is proposed, which allows taking into account the acute oral and chronic toxicity of the pesticide for mammals and its impact on the environment (half-life in soil and chronic toxicity for aquatic organisms), as well as the rate of application of the drug. The computation was performed on fungicides and insecticides that are most commonly used in grain crop cultivation and approved to be applied in the Russian Federation. The research was carried out in 2022–2023. The results showed that the determined indicator takes values from 0.02 for the fungicide benomyl to 26950 for the insecticide chlorpyrifos. Pesticides were ranked according to the $H_r$ index, and four hazard groups were identified, as well as the main factors that determine them. The first hazard group should include drugs with a $H_r$ greater than 1000; the second hazard group—with $H_r$ from 100 to 1000; the third—with $H_r$ from 1 to 100; and the fourth—with $H_r$ less than 1. The first group includes pesticides with multiple adverse safety profiles, such as chlorpyrifos. The second group includes insecticides fipronil, lambda-cyhalothrin, gamma-cyhalothrin, imidacloprid and the fungicide flutriafol. The indicator can be used for agro-ecological substantiation of the choice of pesticides for the treatment of agricultural crops and for the selection of priority pesticides for regular monitoring of their content in the environment—primarily the first and second hazard groups. It can be performed remotely by appropriate detectors and sensors. All data about farm pollution can be monitored using an information server monitoring system, the architecture of which has been proposed.

**Keywords:** pesticides; ecological and toxicological assessment; classification of pesticides; pesticide hazard index; pesticide load; data monitoring

## 1. Introduction

Currently, high and stable yields of farm crops cannot be obtained without widespread use of chemical plant protection products against harmful organisms. Being highly efficient

in the control of pests, pesticides are associated with serious risks for the environment and dangers to humans and beneficial flora and fauna. Such pesticide properties such as high toxicity, persistence in the environment, the ability to be transported along food chains and migrate to adjacent environments (water bodies, soil and air), as well as the ability to accumulate in products and, along with constant application, result in the appearance of resistant forms of harmful organisms, require justification, strict regulation of their use in agriculture, and constant pollution monitoring of agroecosystems. It can be performed with the help of information technology.

The pesticide usage regulation is based on a quantitative assessment of their danger to people and natural systems. Assessment of pesticide risks for humans and warm-blooded animals is carried out according to toxicological and hygienic criteria, taking into account such indicators as average lethal doses if injected or in the case of skin contact, average lethal concentration in the air, characterizing oral, skin-resorptive and inhalation substance toxicity, respectively, the functional cumulation coefficient, pesticide resistance in the soil (decomposition time into non-toxic components), as well as the presence of specific effects: mutagenic, carcinogenic, teratogenic, embryotoxic, gonadotropic and allergenic. These toxicological and hygienic criteria, in turn, underlie the development of hygiene standards: the permissible daily dose of a substance for humans and the maximum permissible concentration of pesticides in farm products and environmental objects. However, in order to protect the ecosystem components, beneficial fauna and flora, sanitation and hygiene criteria are not enough; therefore, for a comprehensive assessment of environmental pesticide risks, it is necessary to use ecological and toxicological principles, which include, in addition to toxicity indicators for warm-blooded animals and soil persistence, indicators that take into account pesticide behavior in the environment and their impact on non-target organisms.

Sokolov M.N. and Strekozov B.S. [1] proposed an approach consisting of scoring each of the eleven indicators offered by them according to the developed scales and determining the hazard class of pesticides (three classes in total) by the total score. According to Vasiliev V.P., Kavetsky V.N. and Bublik L.I. [2], this approach, despite a large number of indicators taken into account, is not able to adequately reflect pesticide risks for human health. They projected to assess pesticide threat on the basis of four environmental and toxicological indicators, two of which characterize pesticide risk to humans to the greatest extent (category A)—that is, the median lethal dose of acute oral toxicity for mammals ($LD_{50}$) (the main indicator) and the coefficient of functional cumulation ($C_{cum}$). The other two indicators underline pesticide danger to the environment (category B), that is, soil persistence ($T_{50}$— half-life of non-toxic components) (the main indicator) and the average lethal concentration for fish ($LC_{50}$). On the basis of these criteria, an ecological and toxicological classification of pesticides was developed, including four hazard classes and a method for assessing the danger of pesticide use and predicting ecosystem pollution under specific soil and climatic conditions. Melnikov N.N. proposed to determine the relative risk degree ($E$) of using a particular pesticide, taking into account its application rate, by the formula [3]:

$$E = P \cdot \frac{T_{50}}{LD_{50}},\qquad(1)$$

where $P$ is the application rate of the drug for the active substance, kg/ha or g/ha;

$T_{50}$ is a half-life in the soil for non-toxic components, weeks;

$LD_{50}$ is the median lethal dose of acute oral toxicity for mammals, mg/kg.

There are also some approaches to evaluating pesticide risks based on their toxicity to non-target indicator species [4]. Due to the high pesticide toxicity to aquatic organisms, methods for assessing their hazard only to aquatic organisms have been put forward based on their ranking in terms of the average lethal concentration of acute exposure, the maximum inactive concentration (NOEC) and the bioaccumulation factor [4–6].

Some approaches to determining pesticide risks account for both indicators of toxicity, persistence, bioaccumulation, and the predicted substance concentration in the environment or consumption level [5–8]. Mathematical models for predicting the pesticide concentration

in surface water bodies [5,6] and soil have been used to assess the degree of pesticide risk for soil, air, surface and ground waters [9].

Despite the existing methodological diversity, contemporary approaches to measuring pesticide risks are characterized by weak convergence between them and an insufficient degree of complexity [10–13]. The available data on hazardous properties of pesticides, contained, for example, in the PPDB (Pesticide Property Database) [14], makes it possible to search and select indicators to use them for an integral toxicological and hygienic assessment of pesticide hazard. One of the factors that determine pesticide risks for the environment is the pesticide application rate [1–3,9]. Moreover, approaches based on risk assessment of pesticide application include pesticide content in the environment as one of the main indicators [5–7], which largely depends on the drug dose used. Information on the application rates of pesticides can be found in special catalogs or reference books. In Russia, this is the state catalog of pesticides and agrochemicals approved for use on the territory of the Russian Federation [15]. Therefore, technologies that allow reducing the application rates of pesticides without loss of effectiveness against pests are promising; for example, the use of pesticides with silicon compounds [16,17]. Currently, information technologies are increasingly being used in agro-ecological monitoring [18,19]. It has largely become available due to the development of LPWAN/LoRaWAN wireless local area networks [20–23], which allow transmitting large amounts of information over long distances using autonomous power sources (batteries) for 5–7 years. This makes it possible to introduce Internet of Things technology into the agro-ecological monitoring system [24,25], based on the widespread use of all kinds of detectors and sensors [26–29], and to exclude human participation in the process as much as possible, ensuring health and safety. Timely detection of an increased concentration of pesticides in the soil allows for measures to reduce their concentration. Currently, it is proposed achieve this by biological means of protection, such as, for example, the enzymatic hydrolysis of organophosphorus compounds in the soil [30] or the application of exogenous organic matter [31].

The aim of the work was a comparative ecological and toxicological assessment of the danger of pesticides using the value of a complex indicator that takes into account both the toxicity of the pesticide to humans (toxicological criteria) and its impact on the environment (environmental criteria). The indicator can allow one to rank pesticides according to the degree of danger of their use and identify the most dangerous groups that require constant monitoring in environmental objects.

## 2. Materials and Methods

To assess the comparative environmental and toxicological hazard of pesticides, we propose to use the integral indicator of relative hazard, which is a modification of the indicator proposed by Melnikov N.N. [3], taking into account the danger of the pesticide for aquatic organisms, which was proposed by Vasiliev V.P., Kavetsky V.N. and Bublik L.I. [2], since aquatic organisms are among the most sensitive organisms to the action of pesticides. Unlike Vasiliev V.P., Kavetsky V.N. and Bublik L.I. [2], we propose to use not the average lethal concentration for fish ($LC_{50}$) but the NOEC indicator for the most sensitive group of aquatic organisms, which characterizes the risk of chronic exposure.

We also propose that, in addition to the acute toxicity of the substance for mammals, its danger should also be taken into account in the event of a long-term intake of the substance into the body, which can manifest itself, among other things, with long-term specific effects. The most readily available indicator, established for all pesticides, that should reflect this hazard to humans is the acceptable daily intake (ADI), established on the basis of thresholds for chronic action and possible long-term effects.

When determining the integral indicator of the relative ecological and toxicological hazard ($H_r$), we propose to take into account the chronic toxic effect on mammals (in terms of ADI) and aquatic organisms (in terms of NOEC) using the coefficients $K_{ADI}$ and $K_{NOEC}$, respectively. Suggested values of the coefficients $K_{ADI}$ and $K_{NOEC}$ are presented in Table 1. The values of the coefficients are taken depending on the values of the indicators ADI and

NOEC: the lower the ADI and NOEC, the higher the $K_{ADI}$ and $K_{NOEC}$, respectively. Thus, for pesticides with an ADI of 1 mg/kg or more, the $K_{ADI}$ is taken to be equal to 0.5; for pesticides with high toxicity to mammals, the ADI of which is thousandths of a milligram, the $K_{ADI}$ is taken to be equal to 3, ten-thousandths—4. When the ADI of a pesticide is tenths or hundredths of a milligram, the $K_{ADI}$ is taken to be equal to 1 and 2, respectively. When the NOEC value for the most sensitive group of aquatic organisms is tenths and hundredths of a milligram per liter, $K_{NOEC}$ is taken to be equal to 0.5 and 1, respectively; when the NOEC value is hundredths and thousandths of a milligram per liter, 2 and 3, respectively; and hundred-thousandths and millionths of a milligram of a substance per liter, 4 and 5, respectively (Table 1).

**Table 1.** Proposed hazard coefficients $K_{ADI}$ and $K_{NOEC}$, taking into account chronic toxicity for mammals and aquatic organisms.

| ADI, mg/kg | $K_{ADI}$ | NOEC, mg/L | $K_{NOEC}$ |
|---|---|---|---|
| 1 and more | 0.5 | A tenth | 0.5 |
| A tenth | 1 | Two decimal places | 1 |
| Two decimal places | 2 | Three decimal places | 2 |
| Three decimal places | 3 | Four decimal places | 3 |
| Four decimal places and less | 4 | Five decimal places | 4 |
| | | Parts per million | 5 |

The relative degree of risks associated with pesticide usage, or relative hazard index ($H_r$), can thus be estimated by the formula:

$$H_r = P \cdot \frac{T_{50}}{LD_{50}} \cdot K_{ADI} \cdot K_{NOEC}, \tag{2}$$

where $P$ is the application rate of the drug for the active substance, g/ha;

$T_{50}$ is the half-life in the soil for non-toxic components, days;

$LD_{50}$ is the median lethal dose of acute oral toxicity for mammals, mg/kg.

$K_{ADI}$ is a coefficient that takes into account the chronic toxic effect on mammals and is taken depending on the ADI value;

$K_{NOEL}$ is a coefficient that takes into account the chronic toxic effect on aquatic organisms and is taken depending on the NOEC value for the most sensitive group of aquatic organisms.

Since a significant portion of pesticides currently have a shorter time of degradation in the environment, we suggest that the half-life in the formula be given in days.

The relative danger of using one or another pesticide will depend both on the rate of its application per hectare for the active substance and on its specific relative hazard index (*$H_r(sp)$*), determined by the ratio:

$$H_r(sp) = \frac{T_{50}}{LD_{50}} \cdot K_{ADI} \cdot K_{NOEC} \tag{3}$$

We have calculated the indicators of relative environmental and toxicological hazards of fungicide and insecticide application that are generally used in grain crop cultivation and approved for usage in the Russian Federation [15], their ranking and their grouping. The research was carried out in 2022–2023. The data source on pesticide properties ($T_{50}$, $LD_{50}$ and NOEC) was the PPDB (Pesticide Property Database) [14].

## 3. Results and Discussion

The results showed that the determined indicator $H_r$ takes values from 0.02 for the fungicide benomyl to 26,950 for the insecticide chlorpyrifos (CPS). Pesticides were ranked

according to the $H_r$ index, and four hazard groups were identified, as well as the main factors that determine them (Table 2). The first hazard group should include drugs with a relative hazard index ($H_r$) of more than 1000, the second hazard group—with $H_r$ from 100 to 1000, the third—with $H_r$ from 1 to 100 and the fourth—with $H_r$ less than 1.

**Table 2.** Indicators of the relative environmental and toxicological hazards of insecticides.

| Insecticide | Application Rate, g Active Substance/ha | $C_{ADI}$ | $C_{NOEC}$ | Specific Hazard Index, ($H_r$ (sp)) | Hazard Index ($H_r$) | Hazard Group |
|---|---|---|---|---|---|---|
| Chlorpyrifos (PT) | 384 | 3 | 4 | 70.2 | 26950 | 1 |
| Fipronil (PT) | 24 | 4 | 3 | 18.5 | 444 | 2 |
| Lambda-cyhalothrin (PT) | 7.5 | 3 | 5 | 47.0 | 352 | 2 |
| Gamma-cyhalothrin (PT) | 36 | 3 | 5 | 7.31 | 263 | 2 |
| Imidacloprid (ST) | 60 | 1 | 2 | 2.86 | 171 | 2 |
| Imidacloprid (PT) | 49 | 1 | 2 | 2.86 | 140 | 2 |
| Clothianidine (ST) | 35 | 1 | 2 | 2.18 | 76.3 | 3 |
| Cypermethrin (PT) | 75 | 2 | 4 | 0.62 | 46.2 | 3 |
| Clothianidine (PT) | 17.5 | 1 | 2 | 2.18 | 38.2 | 3 |
| Alpha-cypermethrin (PT) | 10 | 2 | 3 | 3.51 | 35.1 | 3 |
| Fenitrotion (PT) | 400 | 3 | 3 | 0.07 | 29.5 | 3 |
| Tau-fluvalinate (PT) | 48 | 3 | 4 | 0.61 | 29.3 | 3 |
| Thiamethoxam (PT) | 175 | 2 | 1 | 0.16 | 27.1 | 3 |
| Beta-cypermethrin (PT) | 10 | 3 | 3 | 2.62 | 26.2 | 3 |
| Dimethoate (PT) | 400 | 3 | 2 | 0.06 | 24.5 | 3 |
| Deltamethrin (PT) | 7.5 | 1 | 5 | 1.62 | 12.2 | 3 |
| Malathion (PT) | 285 | 2 | 3 | 0.001 | 0.25 | 4 |

Note: (PT)—plant treatment; (ST)—seed treatment.

Of the considered insecticides, the greatest danger is the use of CPS, whose indicator of relative environmental and toxicological hazard ($H_r$) is very high—it is many times higher than that of other pesticides. This is due both to its high application rate and its high specific relative hazard—70.2, due to long-term persistence in soil ($T_{50}$ greater than one year) and high toxicity both for mammals (acute $LD_{50}$—66 mg/kg, ADI—0.001 mg/kg) and for aquatic organisms (NOEC for the most sensitive group of aquatic organisms (invertebrates)—0.0004 mg/L). It belongs to the first group of dangers.

Fipronil, lambda-cyhalothrin, gamma-cyhalothrin and imidacloprid have high values of the $H_r$ index—444, 352, 263 and 171 (imidacloprid spraying of plants), 140 (imidocloprid seed treatment), respectively. They can be assigned to the second hazard group ($H_r$ from 100 to 1000). At the same time, lambda-cyhalothrin and fipronil have higher specific relative hazard values—47.0 and 18.5, respectively, ranging from 10 to 50. In lambda-cyhalothrin, this is due to its high toxicity to aquatic organisms—the NOEC for the most sensitive group of aquatic organisms (invertebrates) is 0.0000022 mg/L and for fipronil with a high chronic toxicity value for humans—ADI—0.0002 mg/kg, while both are quite persistent in the soil (half-life is more than 6 months). Specific hazard indicators ($H_r$(sp)) for gamma-cyhalothrin and imidacloprid are much lower, at 7.31 and 2.86, respectively.

The remaining insecticides have $H_r$ values less than 100—they can be assigned to the third hazard group, with the exception of malathion, which is assigned to the fourth group ($H_r$ less than 1). The most unfavorable properties among insecticides in the third group are characterized by alpha-cypermethrin, beta-cypermethrin and clothianidin, which have specific relative hazard indicators of 3.51, 2.62 and 2.18, respectively. In alpha-cypermethrin,

this is primarily due to its high acute toxicity to mammals ($LD_{50}$—40 mg/kg). Beta-cypermethrin is less toxic to mammals ($LD_{50}$ is 93 mg/kg), but more toxic to aquatic organisms—the NOEC for the most sensitive group of aquatic organisms (chironomids) is 0.00006 mg/L. Among the considered insecticides of the third group, the most toxic for aquatic organisms are cypermethrin, tau-fluvalinate, and, especially, deltamethrin, whose NOEC for the most sensitive group of hydrobionts is 0.00003 (fish), 0.000024 (invertebrates) and 0.0000041 (invertebrates) mg/L, respectively.

Among insecticides in the third group, dimethoate, fenitrothion, and thiamethoxam have low indicators of specific relative hazard—0.06; 0.07; and 0.16, respectively. Dimethoate and fenitrothion decompose very quickly in the soil ($T_{50}$ is less than three days); thiamethoxam decomposes much longer in the soil ($T_{50}$ is 121 days) but is less toxic to mammals and aquatic organisms. Their inclusion in the third group is largely due to their high application rates. The insecticide malathion for aquatic organisms is even more dangerous than dimethoate and thiomethoxam, but in general it is characterized by lower $H_r$ values since it decomposes very quickly—within a few hours—in the soil and has low toxicity to humans.

Most fungicides are less dangerous than insecticides, especially for insects (hazard classes for bees are set separately). This is also true for aquatic organisms: only chlorothalonil has a $K_{NOEL}$ of 2. None of the considered fungicides belongs to the first hazard group, and only flutriafol can be attributed to the second group (Table 3). With low toxicity to mammals and aquatic organisms, it has a very long half-life in soil—1587 days.

**Table 3.** Indicators of the relative ecological and toxicological hazards of fungicides.

| Fungicide | Application Rate, g Active Substance/ha | $C_{ADI}$ | $C_{NOEC}$ | Specific Hazard Index, ($H_r$ ($sp$)) | Hazard Index ($H_r$) | Hazard Group |
|---|---|---|---|---|---|---|
| Flutriafol (PT) | 125 | 2 | 1 | 2.78 | 348 | 2 |
| Ciproconazole (PT) | 60 | 2 | 1 | 0.81 | 48.7 | 3 |
| Triadimephone (PT) wheat | 250 | 2 | 1 | 0.17 | 43.3 | 3 |
| Triadimephone (PT) barley | 125 | 2 | 1 | 0.17 | 21.7 | 3 |
| Propiconazole (PT) | 125 | 2 | 0.5 | 0.13 | 16.3 | 3 |
| Tetraconazole (PT) | 100 | 3 | 0.5 | 0.12 | 12.2 | 3 |
| Pentiopyrad (PT) | 200 | 1 | 1 | 0.06 | 12.2 | 3 |
| Triticonazole (PT) | 40 | 2 | 1 | 0.25 | 9.84 | 3 |
| Epoxiconazole (PT) | 75 | 1 | 1 | 0.11 | 8.39 | 3 |
| Fluxapiroxad (PT) | 33.3 | 2 | 1 | 0.18 | 6.09 | 3 |
| Chlorothalonil (PT) | 1250 | 2 | 2 | 0.003 | 3.53 | 3 |
| Tebuconazole (ST) | 6 | 2 | 1 | 0.43 | 2.58 | 3 |
| Carbendazim (PT) | 250 | 2 | 1 | 0.007 | 1.72 | 3 |
| Tiram (ST) | 240 | 2 | 1 | 0.005 | 1.30 | 3 |
| Carbendazim (ST) | 100 | 2 | 1 | 0.007 | 0.69 | 4 |
| Metrafenone (PT) | 30 | 1 | 0.5 | 0.02 | 0.60 | 4 |
| Fludioxonyl (PT) | 10 | 1 | 1 | 0.04 | 0.44 | 4 |
| Benomil (PT) | 250 | 1 | 1 | 0.0001 | 0.02 | 4 |
| Benomil (ST) | 200 | 1 | 1 | 0.0001 | 0.02 | 4 |

Note: (PT)—plant treatment; (ST)—seed treatment.

The fourth group includes benomyl, metrafenone, fludiaxonil and carbendazim used for seed treatment. When spraying plants, due to a higher application rate, the indicator of relative danger also increases, which gives reason to attribute such use to the third hazard group. The rest of the considered fungicides also belong to the third group. Fungicides, having less toxicity than insecticides, decompose in the soil for a long time—$T_{50}$ is more than six months for such fungicides as metrafenone, fludiaxonil, fluxopiroxad, triticonazole, tebuconazole, cyproconazole and epoxiconazole.

If we compare the indicators of environmental and toxicological hazards of pesticides obtained by us in the calculation with the results of the calculation according to the formula proposed by Melnikov N.N. [3], then the differences will be as follows. The differences in the accepted values of the indicator for different pesticides when using the indicator proposed by us are greater, which is associated with taking into account a larger number of factors. In addition, taking into account the indicator of chronic toxicity for mammals and aquatic organisms significantly affects the values of the relative hazard of pesticides. So, for example, comparing the hazard indicators of gamma-cyhalothrin and deltamethrin using the work by Melnikov N.N. [3], we find that the relative hazard index of gamma-cyhalothrin is 7.5 times greater than that of deltamethrin, and the specific hazard index is 1.5 times greater, using our indicator, 21.6 and 4.5 times, respectively. This was influenced by taking into account chronic toxicity for mammals, which differs for these drugs and is taken into account by the $K_{ADI}$ coefficient: for deltamethrin, it is taken to be equal to 1, for gamma-cyhalothrin—3 (with close half-lives in soil—28.2 and 26.8, respectively). When comparing the relative danger of imidacloprid with the danger of fipronil, lambda-cyhalothrin and gamma-cyhalothrin according to the criterion of Melnikov N.N., imidacloprid has a higher danger: its rate is two times greater than that of fipronil and approximately four times greater than that of lambda-cyhalothrin and gamma-cyhalothrin. Our results show a greater danger of fipronil, lambda-cyhalothrin and gamma-cyhalothrin, which are associated with a significantly higher chronic toxicity of the latter for mammals and especially aquatic organisms, than imidacloprid.

Thus, our indicator allows us to take into account a larger number of factors and, based on them, obtain a ranked series of pesticides with a greater difference in values. Therefore, we recommend using it for agro-ecological justification when choosing a pesticide for the treatment of crops, in particular cereals, and for identifying priority pesticides for regular monitoring of their content in the environment—primarily the first and second hazard groups.

At the same time, in both approaches, both ours and Melnikov's N.N. [3], one of the factors that determine the danger of pesticide use is the dose applied per hectare, which is also consistent with the data of Y. Zhan and M. Zhang (2013) [9], obtained on the basis of a mathematical model analysis to identify indicators highlighting pesticide risks. According to our data (Table 3), the fungicide tebuconazole has a specific hazard index that is seven times higher than that of penthiopyrad, while the hazard index of using penthiopyrade is almost five times higher, which is due to a higher application rate.

In addition, if we take the ratio of the application rate of the pesticide to the median lethal dose of acute oral toxicity for mammals ($P:LD_{50}$), we get the number of mean lethal doses applied per hectare, and the larger it is, the more dangerous the use of the drug, but only in terms of acute toxicity for mammals. This indicator will be greater the higher the application rate and the lower the $LD_{50}$. In the considered insects, this indicator has the maximum value for chlorpyrifos: when using it, 5818 median lethal doses are applied per hectare, which is significantly more than that of dimethoate and fenitrothion following it in this indicator—1633 and 1212, respectively; the lowest rates are for clothianidin, deltamethrin and tau-fluvalinate (less than 100). For all fungicides considered, it is significantly lower than for insecticides (less than 100), while triadimefon has the highest indicator. However, as we noted earlier, in order to characterize the danger of a drug, it is not enough to take into account only its acute toxicity for mammals; it is preferable to use criteria that take into account a larger number of factors. At the same

time, the application rate of the pesticide, which is one of the most important factors, must be taken into account. Therefore, technological methods that allow reducing pesticide application rates without losing effectiveness against pests will also help reduce the risk of their application, including technologies using silicon compounds.

In our previous field of studies, it was found that the use of silicon compounds such as tetraethoxysilane (TES) and sodium silicate in tank mixtures with pesticides increased the effectiveness of the latter and allowed reducing the application rates of fungicides by 50% and insecticides by 20% without reducing the effectiveness of protective agents [16,17]. The decrease in application rates will also affect the indicator of the relative ecological and toxicological danger of use: for fungicides, it will decrease by two times, and for insecticides—by 20%. The use of low application rates of chemical plant protection products is currently one of the directions for searching for the safest pesticides for the environment, along with increasing the selectivity of their action against harmful organisms and reducing resistance in environmental objects.

Limitations and recommendations. We calculated the proposed indicator for insecticides and fungicides, the most commonly used in the cultivation of grain crops and approved for use on the territory of the Russian Federation. However, we can recommend it for use with other pesticides used in other countries since the criteria included in it are known for all registered pesticides and are contained in the public database of pesticide properties, the PPDB (Pesticide Property Database) [14].

The remote method of determining the amount of pesticides using computer monitoring will replace the traditional method. Currently, the procedure for sampling representative soil samples on arable land and in gardens means that the frequency and timing of sampling are carried out in accordance with the requirements of GOST 17.4.3.01 [32]. Sampling at a site located 50 m from the road (forest edges, ravine edges, bushes, field corners) is carried out by two operators with special shoes and a portable support (stick).

The business process "as is" includes the following operations:

1. Operators go to the sampling site;
2. In the central part of each representative site, one test site with a size of $100 \times 100$ or $100 \times 200$ m is laid. On non-arable lands (meadow, field, pasture, fallow, virgin land), when single samples are taken with a shovel, the combined sample consists of five single samples taken by the "envelope" method (one single sample in the corners of the test site and one single sample in the center) to a depth of 0 to 10 cm;
3. The selected single sample is separated from the bayonet of the shovel manually or with a knife. When single samples are taken by a soil drill, the combined sample consists of 20 single samples taken diagonally from the test site to a depth of 0 to 10 cm;
4. Transport, store, and prepare soil samples for analysis in accordance with the requirements of GOST 17.4.3-01;
5. Determination of the mass fraction of the residual amount of pesticides in soil samples is carried out in accordance with the requirements of GOST 17.4.3.03 and RD 52.18.656 -2004 by gas-liquid chromatography [33].

Pesticides are extracted from the soil by extracting them with a mixture of acetone and 0.05% calcium chloride solution, followed by the redistribution of pesticides into hexane.

Identification of pesticides is carried out according to the retention time set using a calibration solution.

The determination of the mass fraction of pesticides in the sample is carried out by the GC method by comparing the height (area) of the peak of the analyzed and calibration solutions.

6. Preparation of a working solution of calcium chloride and sodium chloride;
7. Sampling 98 cm$^3$ of distilled water into a flask, combining with solutions of calcium and sodium and mixing;

8.　From an air-dry sample, a sample is taken by quartering for analysis, weighing from 200 to 300 g. Roots and other foreign particles are carefully removed from the sample for analysis; the soil is ground in a porcelain mortar and sifted through a soil sieve with a hole diameter of no more than 0.5 mm;

9.　A part of the sample for analysis is placed with a spoon or spatula in a bucket; one sample weighing 20 g is weighed in a bucket on a high-precision scale, and the sample is placed through a funnel into a conical flask with a capacity of 250 $cm^3$;

10.　Extraction of pesticides from the sample.

The pesticide content can be determined remotely (the business process "to be"). For example, in the study [18], an example of a fluorescent sensor for the pesticide staran based on bis-tetraphenylimidazole bound by thiourea bridges (TBTPI) was presented.

As was outlined above, the traditional method of determining the level of pesticides in the soil is a multi-step, time-consuming and dangerous to human health process that lasts several days. A new method based on Internet of Things technologies allows you to reduce work time.

Some references to GIS technology applications can be found as well [19]. GIS technology is used to connect with the Internet and establish a monitoring system for the assessment of chemical pesticide pollution based on the rudimentary collected data. The system analyzes the use of pesticides, as well as the areas and trends of use, and presents the results in the form of chart data, providing comprehensive and reliable information for chemical pesticide pollution monitoring. In particular, it will be undertaken using long-range, low-capacity Lorawan networks [20]. Similar proposals have been made by other scientists [21,22].

Thus, soil contamination with pesticides can currently be monitored using information systems. For example, a server system for monitoring information related to the agricultural production environment in the field has been offered [23].

The collected data are converted into a database through the agricultural environment monitoring server, which consists of a sensor manager, which manages information collected from the WSN sensors, an image information manager, which manages image information collected from CCTVs, and a GPS manager. The authors propose the following architecture for the information system (Figure 1) [24]. The top-level protocol of the OSI network model provides interaction between the network and the user—the environmental service of the region. It includes Environment Monitoring Service, Image Monitoring Service, Location Monitoring Service and Situation Notice Service. The information system will have a unified database on the state of all natural objects. This will allow for comprehensive environmental monitoring.

After sensors are distributed outdoors and every sensor node forms an autonomous network, they send physical information acquired from sensor nodes wirelessly to the server system. The sensor manager manages data acquisition from the soil and environmental sensors, extracts the soil and environmental data by processing the collected data packets into a format that could be stored in the database, stores it in the database, or sends it to other server systems for processing by converting the processed data into a format suitable for the measurement elements [25].

At the physical layer of this system, we propose to add a sensor for the presence of pesticides in the soil, which will give a more complete picture of environmental pollution.

There are many publications on environmental monitoring information systems [24–29]. Timely detection of an increased content of pesticide in the soil will allow farm producers to reduce the negative impact on the environment and yield. For example, a method of enzymatic hydrolysis of organophosphorus compounds in the soil was proposed [30]. At the same time, a biocatalyst is introduced into the soil—an uncleaned polyhistidine-containing polypeptide with the properties of organophosphate hydrolase, obtained from a strain of Escherichia coli bacteria CCMIBX 29 and immobilized on a cellulose-containing carrier. The authors of [31] investigate other proposals concerning the solution to this problem.

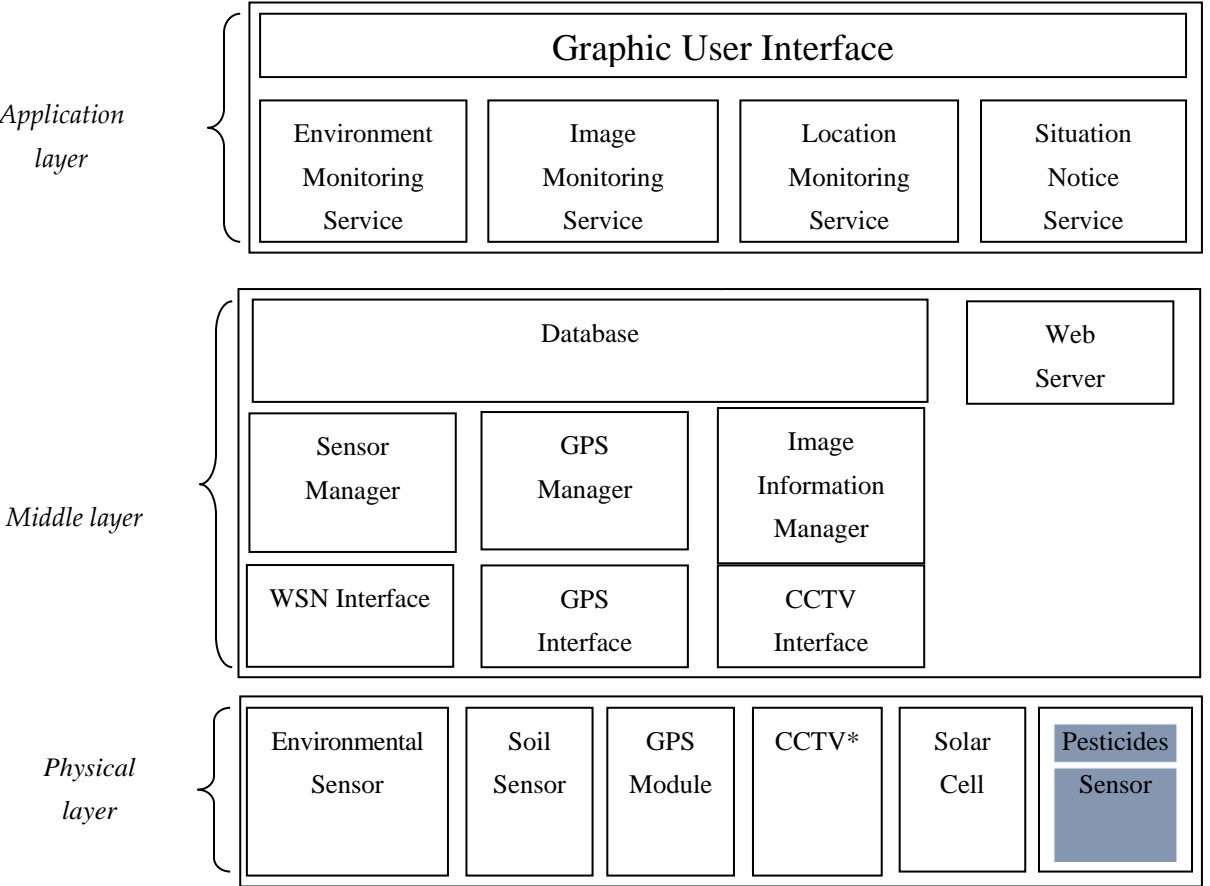

\*  *CCTV-Closed Circuit Television Interface*

**Figure 1.** Agricultural environment monitoring server system architecture to monitor the pesticide content in the soil (developed on the basis of [24]).

## 4. Conclusions

Most of the pesticides currently used are characterized by a number of adverse properties. We propose an integrated indicator of the relative environmental and toxicological hazards of pesticides that takes into account the most important hazardous properties of pesticides, such as acute and chronic toxicity for mammals, chronic toxicity for the most sensitive group of aquatic organisms, half-life in soil and application rate per hectare. The indicator $H_r$ can be calculated for any pesticide registered in the countries for which data on properties are known. We used the open database on the properties of pesticides, PPDB (Pesticide Property Database).

Our calculation of the indicator of the relative ecological and toxicological hazard of the use of pesticides for insecticides and fungicides showed a variation in the values of the indicator being determined: from 0.02 for the fungicide benomyl to 26,950 for the insecticide chlorpyrifos. Pesticides were ranked according to the $H_r$ index, and four hazard groups were identified, as well as the main factors that determine them. The first hazard group should include drugs with a $H_r$ greater than 1000, the second hazard group—with $H_r$ from 100 to 1000, the third—with $H_r$ from 1 to 100 and the fourth—with $H_r$ less than 1. The first group includes pesticides with multiple adverse safety profiles, such as CPS. The use of such pesticides should be limited, and they should, if possible, be replaced by less hazardous ones. Pesticides of the second group, such as insecticides fipronil, lambda-cyhalothrin, gamma-cyhalothrin, imidacloprid and fungicide flutriafol, should be the object of constant monitoring in environmental objects since they, as a rule, also have several unfavorable environmental and toxicological effects.

The proposed indicator can be recommended both for agro-ecological substantiation of the choice of pesticides for the treatment of agricultural crops and for the selection of priority pesticides for monitoring their content in the environment—primarily the first and second hazard groups. It can be performed remotely by appropriate detectors and sensors. All data about farm pollution can be monitored using an information server monitoring system, the architecture of which has been proposed.

Research on the development of integrated indicators of pesticide risks should be continued to find out the most significant parameters for assessing and unifying researchers approaches, as well as to develop international evaluation criteria. In order to carry out the work more effectively, it is necessary to develop an information system for agro-ecological monitoring.

Currently, in solving the problems of agriculture and environmental safety, digital technologies allow operations to be performed more quickly and at a lower cost. There are new methods based on Internet of Things technologies that allow monitoring the amount of pesticide in the soil. In the context of the integration of information and the creation of a digital ecosystem of agriculture, remote technologies for determining the amount of pesticides in the soil should become part of this system.

**Author Contributions:** Conceptualization, I.S. and E.K.; methodology, I.S. and E.K.; software, M.N.; validation, I.S. and M.S.; formal analysis, M.S. and V.V.; investigation, D.B.; resources, I.V.; data curation, I.S.; writing–original draft preparation, I.S. and T.F.; writing–review and editing, I.S.; visualization, D.B.; supervision, D.B.; project administration, I.S. and E.K.; All authors have read and agreed to the published version of the manuscript.

**Funding:** This research received no external funding.

**Institutional Review Board Statement:** Research does not require ethical approval.

**Data Availability Statement:** The initial data used for calculations can be found in available data on hazardous properties of pesticides, contained, for example, in the PPDB (Pesticide Property Database).

**Acknowledgments:** The article was made with the support of the Ministry of Science and Higher Education of the Russian Federation in accordance with agreement No. 075-15-2022-317 dated 20 April 2022, on providing a grant in the form of subsidies from the Federal Budget of the Russian Federation. The grant was provided for state support for the creation and development of a world-class scientific center "Agrotechnologies for the Future".

**Conflicts of Interest:** The authors declare no conflict of interest.

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
