# Peer review of "Improvement of the Integral Indicator of the Ecological and Toxicological Assessment of the Danger of the Use of Pesticides in Agriculture and the Creation of an Information System for Their Monitoring"

_agriculture, doi:10.3390/agriculture13091797_

Round 1

Reviewer 1 Report (New Reviewer)

I have carefully reviewed your research paper titled "Ecological and toxicological assessment and data monitoring of pesticide application in agriculture," with a particular emphasis on fungicides and insecticides used in grain crop cultivation in the Russian Federation. The study also includes the development of a monitoring system using detectors and sensors." and would like to provide you with some comments:

Major Comments

Lack of Context in Introduction:

The introduction provides a general overview of the subject but lacks a clear statement of the specific research questions, objectives, and hypotheses. It would be beneficial to explicitly define the research gaps and the unique contributions of the current study.

Methodological Details and Clarity:

In the "Materials and Methods" section, some critical details are missing or unclear. It would be beneficial to provide more information about the selection of fungicides and insecticides, the rationale behind the chosen models and equations, and the statistical methods used. There are inconsistencies in the units used for some variables (e.g., T50 in weeks and days). This needs to be harmonized throughout the text.  The description of remote methods, including the use of GIS technology, is not well-integrated into the methodology. More explanation is needed on how these tools were applied and their relevance to the study.

Results and Discussion

                The results and discussion section appears to be fragmented and lacks a cohesive flow. It's essential to present the findings systematically and connect them to the research questions or objectives. Some comparisons are made with other methods or criteria (e.g., Melnikov N.N.), but there's a lack of critical analysis and context to understand why these comparisons are significant.

Conclusion

                The conclusion seems quite brief and does not adequately summarize the main findings, implications, and contributions of the study. Recommendations for future research are mentioned, but they could be elaborated further to guide subsequent investigations in this field.

Language and Formatting

There are some grammatical errors and awkward phrasing throughout the text that need to be addressed.

                The article mentions various technological aspects like Internet of Things technology, GIS, and wireless networks, but the integration of these technologies into the study's framework is unclear. A more detailed explanation of how these technologies were employed and their impact on the study's findings would strengthen the paper.

Citations: Make sure all the references are correctly formatted and all the in-text citations correspond to the right references.

I appreciate the efforts put into this study and encourage you to consider the suggested improvements to further strengthen the scientific impact of your work.

Author Response

According to remark 5 we did not understand where the lines with such numbering are located

Reviewer 2 Report (Previous Reviewer 3)

The revised manuscript is now in a good shape with convincing discussion and relevant improvement of the contents.

However, I suggest adding a comparative discussion in form of a table or figure to focus on the innovations, improvement, and key hypothesis of this new approach for pesticide monitoring as compared to traditional pesticide monitoring thorugh field incurred data. 

I suggest to improve the conclusion with specific recommendations for future studies for assessment of feasibility of this curret approach for pesticide monitoring.

Minor revision is recommended before its acceptance.

Author Response

According to remark 5 we didn't understand exactly where lines 193-212 are located.

Reviewer 3 Report (Previous Reviewer 1)

Accepted

Minor

Author Response

Hello. We have taken into account all your comments

Round 2

Reviewer 1 Report (New Reviewer)

Dear Authors,

I have carefully reviewed your revised manuscript, and I am pleased to see that you have addressed the concerns and suggestions that were raised in the initial review. The improvements made in the manuscript are commendable.

I believe that the revised manuscript has met the standards of quality and rigor expected for publication. The improvements made have addressed the initial concerns, and the paper makes a valuable contribution to the field of ecological and toxicological assessment in agriculture.

I recommend the manuscript for acceptance without further revisions.

Best Regards,

The revised manuscript has improved in terms of language, clarity, and presentation.

This manuscript is a resubmission of an earlier submission. The following is a list of the peer review reports and author responses from that submission.

Round 1

Reviewer 1 Report

1.      I recommend including the abstract, general description of methods, treatments, or evaluations, main results expressed with values and statistical significance, and the conclusion of the evaluation or analysis of the experimental results.

2. Authors insert a reference for this formula consumption rate (Co).

3. Insert the references in all formula of the manuscript.

4. Author need to write the hypothesis at the end of the introduction.

5. from number 193 to 197 the font must modified to the style of journal.

6. The discussion is unsuitable to publish, you must focus on your work by ‎discussing your results step by step and some of the citations remove from ‎the discussion is suitable to mention in the section introduction.

7. The conclusions are weak.

8. Extensive editing of English language required

Extensive editing of English language required

Reviewer 2 Report

1. Explain the meaning of each parameter of equation 2, and where the values were extracted to make the calculations.

2. What criteria were taken into account in table 2 to assign the values of CADI, CONEC.

3. The methodology should be explained better, it is not clear.

4. Make a complete and clear discussion of the results obtained in this investigation with the one obtained by V.P.Vasiliev, V.N. Kavetsky, and L.I. Bublik.

5. The description located on lines 193 to 212 with the current investigation is not clear. Please, explain better figure 1.

Reviewer 3 Report

The manuscript titled “Logical and Toxicological Assessment and Data Monitoring of Pesticide Application in Agriculture” is reviewed and found to be inappropriate for publication in its current form. The manuscript is trying to establish a database monitoring system through sensors and IoT servers. However, the novelty, data arrangement, and figures/tables are not enough to satisfy the publication criteria. The following are the specific points for further improvement while this paper needs to be reorganized with focused findings and can be resubmitted in this journal or elsewhere.

1.      The title should be catchier with focused novelty. The abstract should include the following sequence: objectives, methods, results, and recommendations. What is new/novel for this manuscript should be mentioned properly?

2.      The methodology should be reorganized with IoT and traditional field-based monitoring of pesticides. This kind of contrasting methodology is highly encouraged.

3.      The results and discussion section of this manuscript is poor with no specific recommendations. In general, traditional monitoring is practiced globally for pesticide safety guidelines. What are the limitations of real field pesticide monitoring with alternative better options can be wondrous to make your manuscript vivid with a logical explanation for this adopted technology.

4.      Add a dedicated section “Limitations and recommendations” before the conclusion.

5.      Revise the last paragraph of the Introduction with specific objectives and follow the objectives-based outcomes.

6.      Revise “Conclusion”. Why this study is important over the existing monitoring approaches of pesticide contamination.

Reject and encourage resubmission.

Moderate revision is required.